# An Orientalist Contribution to "Catholic Science": The Historiography of Andalusi Mysticism and Philosophy in Julián Ribera and Miguel Asín

## Pablo Bornstein

The Zvi Yavetz School of Historical Studies, Tel Aviv University, Tel Aviv 6997801, Israel;
pablobornstein@gmail.com

**Abstract:** This article provides a historiographical analysis of the principal works on Andalusi mysticism and philosophy in Spain at the turn of the twentieth century. It portrays the intellectual background in which the Arabist scholars Julián Ribera (1858–1934) and Miguel Asín Palacios (1871–1944) developed their studies, and their particular "presentist" concerns, highlighting how their works and publications on this field cannot be detached from contemporary national debates on religious issues. The contribution of these Orientalist scholars was especially relevant to the transnational movement in defense of a Catholic science. The adherents of this movement sought ways of stressing the compatibility of dogma with the findings of unbiased scientific works, against the perceived attack to religious doctrine they sensed coming from positivist science. The Spanish Orientalists would bring to light the importance of Eastern Christian thought in the development of medieval Muslim theology, therefore vindicating the Christian origins of Andalusi philosophical and theological production and rendering it easier for the Catholic Spanish public to come to terms with Orientalist queries.

**Keywords:** Orientalism; al-Andalus; mysticism; scholasticism; Catholic science; Spain; historiography

---

## 1. Introduction

In the fall of 1893, the Arabist scholar Julián Ribera (1858–1934) addressed the students and teaching staff of the University of Zaragoza in the inaugural lecture opening the academic year, a common practice at the Spanish universities in the nineteenth century. Ribera chose as the topic for his lecture the history of the educational practices and institutions among Andalusi Muslims. What made his address remarkable was the explanation that Ribera provided in order to justify the interest and relevance of his chosen subject matter. At a moment of international decline of Spain, in which the idea of national "regeneration" was gaining ground among the Spanish intelligentsia, Ribera claimed that he had decided to engage in that study as a means to analyze "the spirit showed by our race toward the teaching of sciences and arts within a civilization that is so different from the Christian one", with the aim of pondering whether any lesson could be drawn from the Andalusi experience that could serve as a "stimulus" for present-day Spaniards (Ribera 1893, p. 7). Such a dissociation between "race" and "civilization" portrayed an inclusive understanding of Spanish national identity that allowed him to consider the Andalusi Muslims as part of the Spanish "race" and, therefore, as related to some extent to modern Spaniards. This was far from being universally accepted, and such a vision would have been hard to elaborate a century earlier. It was the work of Orientalist scholarship throughout the nineteenth century that allowed for new interpretations of the Muslim legacy to Spain (López García 2011; Marín 2014; Rivière 2000). And that scholarship was not without nationalist purposes, reclaiming for Spain the riches of Andalusi culture. Ribera concluded his lecture affirming that it had been the

"genius of Spain" that touched the Andalusi men of science and accounted for how "our motherland managed to become, through the endeavor and zeal of its offspring, the master of the Western nations" (Ribera 1893, p. 99).

This article analyzes the research pursued at the turn of the twentieth century by Julián Ribera and his Arabist colleague Miguel Asín Palacios (1871–1944) into the philosophy and theology of Andalusi Muslims in light of the contextual realities and intellectual debates of the time, discussions that had a profound impact on their scholarship. Given the leading roles of both scholars in incorporating those type of studies within the body of Spanish Orientalism, the "presentist" concerns embedded in their investigations would have a great weight in the manner in which the Andalusi philosophical and theological legacy would later be considered and portrayed among several generations of scholars. The article highlights the support given by the leading conservative scholar and director of the National Library of Spain Marcelino Menéndez Pelayo (1856–1912) to Ribera and Asín Palacios in their surveys of the intellectual history of al-Andalus. Menéndez Pelayo encouraged them to strive for a better understanding of the influence of Oriental philosophies in the development of medieval Christian thought. I would argue that Menéndez Pelayo came to assume the value of the Orientalist field prompted by the fact that the Arabists' findings were consistent with his program to vindicate the history of Spanish intellectual production. The analysis of the philosophical and theological exchanges between Islam and Christianity received renewed attention at the turn of the century due to the interest taken by Ribera during the 1890s in the educational institutions of the Islamic world, which he connected to the cultural preoccupations felt in Spain at the time and tied to the Regenerationist movement.

Just a few years later, Miguel Asín Palacios would argue that some trends of Islamic medieval thought had a determining influence on the development of Christian scholasticism. Paradoxically, such a claim, which if made some decades earlier would have received the unanimous condemnation of most Catholic men of letters, was raised by a Catholic priest—Asín Palacios had been ordained in 1895. I argue in this article that a process of "nationalization" of the Iberian Semitic heritage carried out by the Spanish Orientalists throughout the nineteenth century had lasting effects, and indeed, at the turn of the twentieth century, the scholars who would make the most dedicated efforts to demonstrate the influence of Iberian Islamic and Jewish thought and culture over Europe were all devoted Catholics of conservative leanings. Their works would even be favorably received in some ecclesiastical intellectual circles. A key to this success was the fact that Ribera and Asín Palacios, as is shown in this article, developed a theoretical framework which explained the whole of medieval Muslim philosophy as stemming from the reception in the East of the intellectual traditions of early Christian thought. Thus, to some extent, Ribera and Asín Palacios would "Christianize" Muslim and Andalusi philosophy, rendering it easier for the Catholic public to come to terms with the Arabist arguments regarding their lasting influence in Spain. This article seeks to expose how the Orientalist studies of Ribera and Asín Palacios and their interpretation of the Andalusi legacy to Spain are inseparable from the goals of a transnational movement that was looking to legitimate a Catholic science within European universities.

## 2. Engaging with the "Internal History" of al-Andalus

### 2.1. The Emergence of "The Arabist School"

Arabist scholarship in Spain experienced an unprecedented development throughout the nineteenth century. Spurring that development was the unstoppable spread of Spanish nationalism, which fostered a new preoccupation with all the cultural phenomena that had once taken place in Spain. Spanish reluctance to engage with the Iberian Semitic past, if not outright suppression of it, began to erode gradually, as a wider range of scholars considered the impact of the legacy left by Jews and Muslims to Spanish culture and mores. The great works of Iberian Muslim and Jewish thinkers and men of science were nationalized, praised, and glorified as forming part of the cultural wealth of the Spaniards, and surviving Arab manuscripts that had for centuries been ignored were incorporated into the study of the Spanish past in the belief that they could bring new light to various episodes



of the country's history. Several generations of scholars, among them figures such as José Antonio Conde (1766–1820), Pascual de Gayangos (1809–1897), José Amador de los Ríos (1818–1878), Francisco Javier Simonet (1824–1897), Francisco Fernández y González (1833–1917), and Francisco Codera (1836–1917), had succeeded in generating a wide-ranging debate at the national level concerning the inclusion of Muslim and Jewish letters within the national canon (López García 2011). The result was a "Hispanicization" of the Semitic past, instrumentalized as a means to vindicate the cultural greatness of the nation and made at a time of widespread sense of national decline. However, the religious affiliation of those medieval Iberian Jews and Muslims did not cease to represent an inconvenient obstacle for the complete absorption of that cultural legacy within a country in which Catholicism, even among liberals, was still considered to be an underlying trait of the "Spanish character". Towards the end of the century, Orientalist scholarship became more deeply entangled in pressing national discussions on the role that religion should play in contemporary Spain.

Since the 1890s, most scholars familiar with the Arabist field have held that Francisco Codera's success in forming a research group represented a turning point in the course of Spain's Orientalist studies because of the coherence of their aims and the methods they employed. The group, which was basically composed of Codera's students, was informally known as "The Arabist School". In 1899, one of his students in Madrid, Armando Cotarelo y Valledor (1879–1950), published an article praising the virtues of his mentor in a journal directed to university students, *La Juventud escolar* (*The Studious Youth*). In it, Cotarelo included an explicit reference to "the School" initiated by Codera (Cotarelo y Valledor 1899, pp. 23–24). He argued that, despite the fact that an important number of Arabist scholars had emerged in Spain since the eighteenth century, none of their works had aimed for a common, concerted goal. It was Codera who had managed to organize the Arabist field as a collective enterprise with well-defined objectives. The final purpose of this enterprise was none other than that of striving for "a complete knowledge of our Arab history". To achieve this goal, Codera had devoted himself "with extraordinary patience and unwavering will" to gathering information about that past, exploring archives and libraries and extracting the most useful excerpts from Arab sources (Cotarelo y Valledor 1899, pp. 12–14). In 1904, Codera would himself refer explicitly to the "so-called School of Arabists, which despite having by no means official existence, is alive due to the mutual and intimate union of most of those who dedicate ourselves with real interest to these studies" (Codera 1904, p. 425).

Over time, the members of the "School" would develop a eulogistic self-image about the character of the school, which eventually consolidated into a canonical narrative of its origins (Marín et al. 2009, p. 163). Already in 1902, in an article for *Revista de Aragón* (*Journal of Aragon*), the most distinguished of Codera's disciples, Julián Ribera, claimed that all young scholars with aspirations for a career in the field of Arabist scholarship were related to the school created by Codera (Ribera 1902, pp. 274–77). To explain the bond that united the members of "the School", Ribera granted great importance to the work carried on outside of the university classrooms, such as the editing of the volumes of Codera's *Biblotheca arabico-hispana* (*Arabic-Hispanic Library*), a compilation of translated Arabic manuscripts that Codera had edited since 1882. The experience of Codera's disciples in that enterprise lay behind their initiative to begin in 1897 the publication of the *Colección de Estudios Árabes* (*Collection of Arabic Studies*), precisely with the goal of popularizing topics on Spanish Arab history. Together with the journal *Revista de Aragón*, it would constitute the major forum for the dissemination of the research by young Arabists during the turn of the twentieth century.

The Arabist School received in these years the prestigious support of Marcelino Menéndez Pelayo. The endorsement of the leading figure of Spain's conservative letters boosted the intellectual credibility of the members of the School, while also reinforcing their position in the Catholic camp within the Spanish intellectual panorama. The religious leanings of most of the disciples of Codera made them gravitate toward a movement that was developing in defense of "Catholic science". The Spanish adherents of this movement were joining a transnational campaign that had been on the rise during the last decades of the nineteenth century, and that had its principal aim the defense of the validity of Catholic dogmas against the attacks they perceived coming from the academic world. For this

purpose, the "Catholic scientists" stressed the compatibility of those dogmas with the findings of unbiased scientific works. These Catholic apologists tended to make use of a loose and ambiguous understanding of the term "science". They encouraged the contribution of Catholics to research in modern natural, experimental sciences. However, they were especially inclined to combat some of the philosophical consequences that they saw stemming from modern rationalism, and how it was being translated into the field of humanities. The *bêtes noires* of these scholars, as they saw it, were positivism and evolutionism.[1] Eager to demonstrate the compatibility of scientific findings with Catholic theology, they sought to reconceptualize the philosophical debate around modern science in terms of the traditional discourse of Catholic scholasticism about the relationship between faith and reason (and thus tended to vaguely equate "science" with reason). Rafael Rodríguez de Cepeda, a Spanish champion of this neo-scholastic approach, published a booklet in 1897 explaining to the Spanish public the developments of the international movement for Catholic science. He considered that one of the movement's best achievements was the creation of an institute for the study of Thomistic philosophy in the University of Leuven under the direction of the cardinal Joseph Mercier, whose purpose was to unite "scientific observation to rational speculation in the cultivation of Thomistic philosophy" (Rodríguez de Cepeda 1897, p. 7). After reviewing the alleged success of the centers attached to the movement for Catholic science in "combating the false scientific doctrines" that were being produced in most Western universities, and the contributions they had made both in natural sciences and in the humanities, Rodríguez de Cepeda stressed that "Spain, due to its glorious scientific traditions, must take an active role in the restoration of Christian science" (Rodríguez de Cepeda 1897, pp. 10, 20).

In their efforts to harmonize Catholic faith and reason, by the end of the century some Catholic apologists went on to argue that there was no real contradiction between evolutionist theory and Catholic beliefs. This was the case of the Spanish sociologist Eduardo Sanz y Escartín (1858–1933), who, at a lecture delivered at the Royal Academy of Moral and Political Sciences in 1898, asserted that Catholics who stubbornly condemned the idea of evolution were doing little good to the name of both religion and "science" (Sanz y Escartín 1898, pp. 42–50). He referred his audience to the International Congress of Catholics that had convened in 1894, in which several men of faith had defended the importance of the evolutionist thesis. Sanz argued that many sectarians among Darwin's followers had used his works to attack some of the most sacred beliefs of Christianity, and therefore he saw that it was only natural that good-willed Catholics would come to oppose evolutionism. However, in his opinion, the core principles of the evolutionist theory were not in disagreement with the Catholic dogma, and, indeed, evolutionism was a propitious area for intellectual communication and rapprochement between religion and "science".

In the Spanish case, the defense of Catholic science often merged with a need to defend the national intellectual tradition, which had been the target of European intellectuals for more than a century. As Spanish scholars internalized that criticism, a recurring argument held that the Catholic Church and the Inquisition had exerted suffocating pressure over Spanish minds throughout centuries, that being the main cause for the shortcomings of the Spanish scientific tradition. The controversy intensified in the last years of the 1870s, with some of the most renowned Spanish intellectuals participating in what would become known as the "polemic over the Spanish science" (García Camarero and Camarero 1970). It was in the context of this polemic that a young Menéndez Pelayo obtained nationwide fame as an apologist of the history of Spanish intellectual production, denying that Catholic orthodoxy had had any negative impact on it. These efforts culminated in his work of 1887–1889 *La Ciencia Española* (*The Spanish Science*) (Menéndez Pelayo 1887). At his inaugural address at the Spanish Royal Academy in 1881, which was devoted to Castilian mystical poetry, Menéndez Pelayo made use of the same

---

1   For Catholic strategies of accomodation to modern science, and rejection at the same time of positivist theory of science at the end of the nineteenth century, see: (Motzkin 1989).

ambiguous approach toward the meaning of science that the apologists of Catholic sciences had at the turn of the century. Alluding to the possibility of harmonizing rational speculation and mystical rapture, he invited the public "not to think that *science* is an obstacle for anything; we should not believe, above all, that God's science blocks the way of those who must extoll the divine excellences with the language of rhythm" (Menéndez Pelayo 1881, p. 14). It was at this same event that Menéndez Pelayo pointed to a question that the Arabist disciples of Codera would take as a central issue in their queries: the role of Muslim theologians such as the Persians Avicenna (980–1037) and Al-Ghazali (1058–1111) as cultural intermediaries between ancient Greek philosophy and medieval Christian scholasticism. He argued that "it was not only in the astronomical and physical sciences, but in the very first philosophy that the *sectarians of Islam* served as the chain that connects the ancient culture with the modern" (Menéndez Pelayo 1881, p. 24).

Although the polemic over the Spanish science gradually died down in the following years, derision of Spanish intellectual capabilities emerged anew in the final years of the century, achieving an all-time high after 1898, when the loss of the last remaining imperial territories in Cuba, Puerto Rico, and the Philippines drowned the country in a vast crisis of self-confidence. From his liberal position, the historian Rafael de Altamira (1866–1951) sought to overrun that defeatism in his influential 1902 work *Psicología del pueblo español* (*Psychology of the Spanish People*), asserting that Spain had brought forth illustrious men of science in the past and could very well do the same in the present time (Altamira 1997). Within the conservative camp, Menéndez Pelayo was the Spanish intellectual who had sought to couple, with some success, the efforts to blend Catholic apologetics with the vindication of the so-often vilified Spanish intellectual heritage. It was not accidental then that in 1899, in the aftermath of the Spanish defeat in Cuba, when the doubts about Spaniards' capacities spread all over the country, a group of scholars decided to pay homage to Menéndez Pelayo by organizing a collective volume in his honor. The celebrated novelist Juan Valera (1824–1905) wrote the prologue for the volume, addressing the social malaise that affected Spain in the wake of 1898 (Valera 1899, pp. ix–xxxiv). Struck by the despair voiced by many scholars and intellectuals, Valera—very similarly to Altamira in his *Psicología del pueblo español*—claimed that Spaniards had too easily fallen prey to foreign voices that claimed that Spanish intellectual skills had atrophied under the Inquisition and religious fanaticism. It was true, according to Valera, that Spain was in a difficult situation and that a path to national regeneration was necessary. But first of all, what the circumstances demanded was to put an end to all self-loathing. Noting a change in Menéndez Pelayo from his earlier, more illiberal positions, Valera saw in him the person best suited to restore Spain's lost intellectual confidence. It had fallen upon him the task of "determining, without vagueness and without hesitation, our importance in the history of human thought, and to mark the position that we deserve within the concert of the civilizing nations" (Valera 1899, p. xvii). In the schemes of Menéndez Pelayo, the Arabist school was to play a very relevant part in the task of determining Spain's "importance in the history of human thought". It is therefore not surprising that as many as four Arabist scholars would participate in the homage volume: Francisco Pons (1861–1899), Leopoldo Eguílaz (1829–1906), Julián Ribera, and Miguel Asín Palacios. All of them shared the beliefs and tenets that characterized the scholarly movement for the defense of Catholic science.

Ribera, for example, had praised that same year the alleged support that Catholicism had given to scientific investigations at all historical periods. He delivered a conference paper at the *Ateneo* of Zaragoza on the topic of superstitions among the Spanish *moriscos*, which was later printed in the *Revista crítica de historia y literatura españolas, portuguesas e hispano-americanas* (*Critical Journal of Spanish, Portuguese, and Hispano-American History and Literature*). As a champion of the movement for Catholic science, Ribera broadly identified the notion of science not so much with a positivist methodology based on experimentation but with a general pursuit of rational thought. According to him, Christianity offered a very different picture than Islam with regard to the use of reason, for the Muslim religion had not been able to uproot the old superstitious beliefs from the peoples that it came to dominate. In some cases, the Muslims had even contributed to the spread of those superstitions,

which "fit well with [Islam's] fatalist doctrines and its traditional horror for philosophical studies". Ribera then claimed that, contrary to Islam, Christian dogma was opposed to Manichean dualism and emphasized human freedom. Indeed, he suggested that Christianity had always favored rational thought and, "unfit to pair up with the superstitions of other idolatrous cults, has constantly attempted to suffocate the superstitions upheld by the peoples it converted". Although the Arab invaders had brought their superstitions to Spain with them, the most enlightened sector of Andalusi society, "in spite of Islamism", had devoted itself to scientific inquiries, and had even achieved a better success in them than anywhere else in the Muslim world. However, in his view, the old superstitions had remained intact among the common people, and a good many of them had passed to modern Spaniards, notwithstanding the expulsion of the *moriscos* some centuries earlier. Ribera maintained that the complete uprooting of superstitious beliefs from the masses was an impossible task. But there were two ways to fight superstition: the teaching of Catholic dogma and the study of the human sciences. Catholic men of science should strive to "popularize, especially, those scientific truths that are more accessible to the common man, and do that without outbursts and polemics" (Ribera 1899a, pp. 438–39, 461–62) Ribera's considerations concerning the social utility of science were thus firmly anchored in the program for a Catholic science.

In light of this adherence of Arabist scholars like Ribera and Asín Palacios to the programmatic schemes of the movement for Catholic science, it is striking to realize that these deeply Catholic scholars would indeed be the Spanish intellectuals who went to great lengths in order to study, expose, and call attention to the impact that Iberian Muslim and Jewish philosophers had on the body of Catholic scientific and philosophical thought. Francisco Codera, the venerable mentor of the young Arabists, had refrained throughout his career from engaging in the study of the cultural legacy left by Andalusi thinkers. He considered that the "external" history of al-Andalus, namely its political history, was still incompletely studied, and therefore he stated time and again his desire that Arab scholarship in Spain would limit its investigations to the study of that "external history", until the time was ripe for a different set of analysis on Andalusi "internal history". Now, at the turn of the century, his disciples would open a new avenue of research by questioning what philosophical and religious influences existed between medieval Muslim and Christian authors.

*2.2. The New Agenda of Ribera and Asín Palacios*

It was precisely due to their contribution to the homage volume to Menéndez Pelayo that Codera's disciples first gained notoriety in this area of study. Their projects explored the Muslim influences behind the philosophy of the Majorcan theologian Ramon Llull (1232–1315/16). However, the origins of this inquiry into the impact of Muslim intellectual life on Christian Spain dated back to some years before, when Ribera began to study the educational system in the Muslim world. Later on, Ribera would provide an account of how he had first become interested in the question of the philosophical exchanges and cultural transfers between Christianity and Islam. In this account he explained that, in the course of his investigations into the history of the Muslim educational centers, he had found that an unprecedented level of public involvement in education had taken place roughly simultaneously in separate areas of the Muslim world such as Egypt and Iraq around the end of the tenth century and the beginning of the eleventh century (Ribera 1904, pp. 6–16). After looking into the origins of one of these centers located in Baghdad, Ribera concluded that the new institutions of learning that flourished in that period had taken as a model the *madrasa* that was established by the Persian rulers of Nishapur. The new centers incorporated in their curricula some of the teachings of Eastern monastic sects which, in turn, had been influenced by the doctrines of Eastern Christian schools such as the Nestorians that had amalgamated Christian dogma with ancient Greek philosophy. What his investigations revealed to Ribera was that there had been an uninterrupted series of philosophical exchanges between the Eastern Christian world, the Islamic civilization and, as he and Asín Palacios would unravel, the scholastic

Western Christian tradition.[2]  Ribera affirmed that "the monastic life of the Muslims, and the rules of their religious orders, derived from the Christian forms of the Oriental rites, [and therefore] it is possible to identify the links in the chain that ties all these traditions" (Ribera 1904, p. 16)

In their contribution to the volume dedicated to Menéndez Pelayo, Ribera and Asín Palacios discussed how these links between religious traditions were manifested in the works of Ramon Llull. This was a topic that Menéndez Pelayo himself had considered at a lecture in the *Ateneo* in December of 1898, which he would later reference in a brief article in *Revista Crítica de Historia y Literatura Españolas, Portuguesas e Hispano-Americanas*. In the *Ateneo*, Menéndez Pelayo had argued that Llull could not be considered a canonical scholastic theologian, but rather that he had been "a lonesome thinker, who owed much to the Orient, very little to the Classics, and some to the realism to the Scholastics" (Menéndez Pelayo 1899, p. 80). Ribera and Asín Palacios would delve deeper in the "Oriental" borrowings of Llull. Ribera maintained that, due to the obscure style of the Majorcan theologian, tracing the origins of his thought was one of the biggest challenges in the history of Spanish philosophy (Ribera 1899b, pp. 191–216). He perceived a bizarre structure in Llull's philosophical system which, as Menéndez Pelayo had pointed out, did not resemble that of his contemporary scholastic peers. After arguing that Llull was familiar with Arabic sources and with the works of Muslim Sufi thinkers, he came to the puzzling conclusion that Llull had been indeed a "Christian Sufi". Ribera proposed that both in his life and in his religious opinions, Llull seemed to have been inspired by the Murcian Sufi Ibn 'Arabi (1165–1240). Both had shared similar positions within their respective religions, declaring themselves enemies of exclusively rationalist, irreligious free thinkers yet at the same time feeling alienated from orthodox ones.

The conclusive evidence of Llull's emulation of Ibn 'Arabi was to be found in his *Llibre del Amic y Amat* (*The Book of the Lover and the Beloved*),[3] in which Llull confessed to have found inspiration in a book that he encountered on the Berber Coast that described the activities of local Sufi pious men. Llull's confession was the more striking since he usually avoided mentioning his sources. Thus, Ribera did not hesitate to describe the passage as nothing less than "the point of departure for Spanish Christian Mysticism". According to him, this discovery enabled scholars "to consider new horizons that were never thought of before" and obliged historians of ideas to look closer into the Muslim antecedents of Spanish mystical traditions (Ribera 1899b, pp. 215–16). It also brought onto the radar of Arabist scholars the works of Spanish Muslim mystics like Ibn 'Arabi that had been previously largely unheard of in Europe, despite the fact that his works had achieved more resonance in the Muslim world than other Spanish Muslim authors who were more familiar to the Christian public such as the twelfth-century Andalusi philosophers Ibn Tufail and Averroes.

For his part, Asín Palacios expanded upon the thesis of Ribera, drawing attention to a question at which Ribera had already hinted: the idea that there were, within the works of Muslim thought known to medieval Christian authors, numerous philosophical and theological elements that actually proceeded originally from the intellectual traditions of Eastern Christianity and which would be highly influential in later scholastic models designed to harmonize faith and reason. Asín Palacios warned that the philosophy of Ibn 'Arabi still needed to be studied in more depth, but he was already convinced at that time that Ibn 'Arabi's thought represented a syncretic system combining elements of many Oriental philosophies, in which the ideas of the Alexandrian Neo-Platonist School stood out. The greatest merit of Llull as a philosopher, Asín Palacios claimed, was that, inspired by Ibn 'Arabi's synthesis, he had

---

[2]　A recent account of those philosophical exchanges that underscores—as Ribera and Asín Palacios did at the beginning of the twentieth century—the neo-platonist matrix of many of the dogmas adopted by medieval Christian, Muslim, and Jewish theology and mysticism, can be found in: (Sedgwick 2017), especially the first part: "Premodern Intercultural Transfers." On the origins of Sufism, see: (Karamustafa 2007).

[3]　See a modern edition in (Llull 2015).

introduced into the stream of medieval Christian ideas, purified of its Muslim additives, something that perhaps was nothing else than a Muslim transformation of ancient Christian philosophy. (Asín Palacios 1899, pp. 254–55)

Asín Palacios agreed with Ribera about the potential benefits of this line of investigation. In the first place, it rendered the mystical philosophy of Ramon Llull more comprehensible, which most researchers had previously seen as obscure and mysterious. It also shed light on "an episode in the history of the mystical-pantheistic philosophy of Muslim Spain, whose influence in Islam perpetuated throughout the centuries". But most important of all, this research agenda completely concurred with the goals set by the champions of Catholic science. Given the terms in which Catholic apologists were positing the modern philosophical problem between religion and science as a bid to harmonize faith and reason, the Arabists' new research agenda had the double potential of highlighting "Spanish" contributions to world culture while at the same time stressing important philosophical foundations that could facilitate the goals of Catholic science. Asín Palacios argued that any serious study looking to underscore the relations between Christian scholasticism and Arab philosophy was "a healthy example in these days" (Asín Palacios 1899, p. 255). He appealed to the authority of the encyclical *Aeterni Patris*, issued by Pope Leo XII in 1879, which called for the restoration of Christian philosophy in Catholic schools and for the revival of scholastic thought. Just as Ramon Llull and other medieval scholastics had no shame in adopting from the Muslims what they thought to be suitable for Christian philosophy, Asín Palacios argued that modern Christian scholars should also

take advantage of everything that in the contemporary philosophical literature can be considered legitimate progress, with the firm thought that, in this manner, we will help Christian philosophy to move forward. (Asín Palacios 1899, p. 256)

Asín Palacios thus claimed that such a line of research would breathe new life into the philosophical and theological works that "had turned Spain into, in other centuries, a home for wisdom". Spain, thanks in part to its Muslim philosophers, had once been the center of religious and cultural exchanges that had put forward theological schemes in which rational inquiries could be subordinated to religious doctrines and therefore substantiate religious faith. Asín Palacios's arguments could not have fit better into the cultural project conceived by Menéndez Pelayo for vindicating the intellectual past of Spain. Indeed, Asín Palacios explicitly mentioned how the path that he and Ribera had opened could serve to restore "our glorious scientific traditions, to whose resurrection [Menéndez Pelayo] has dedicated all his initiatives" (Asín Palacios 1899, p. 256).

*2.3. The Backing of Menéndez Pelayo*

In fact, Menéndez Pelayo had played a very significant role in the early career of Miguel Asín Palacios. He was a member of the doctoral tribunal that in 1896 evaluated Asín's doctoral dissertation on the Persian Muslim philosopher Al-Ghazali (c.1058–1111). The personal archive of Asín Palacios conserves some handwritten notes that he took during his thesis defense which bear testimony to the compliments that the professor of the *Universidad Central* bestowed upon the young Arabist scholar.[4] Menéndez Pelayo had expressed his desire that Asín Palacios's thesis be published as a book, and offered suggestions to incorporate in the future publication. Among them was the idea that Asín should explore the ways in which Al-Ghazali's works could have influenced Spanish Christian philosophy. Asín actually referenced this advice in the text that he wrote for the homage volume for Menéndez Pelayo, alluding to how the honoree had prompted him to look for the imprint left by Al-Ghazali on Christian scholastic philosophy. That suggestion had moved him to expand his research so as not only to see the impact on Christian authors but on "Spanish" Muslim philosophers as well, and thus led him to learn about the works of Ibn 'Arabi who had declared himself a disciple of the Persian

---

4    Documentos de la Biblioteca Asín Palacios de la UNED, Caja "Notas e Ideas".

theologian. Asín Palacios considered the works of Al-Ghazali one of the critical links in the chain of the Iberian encounters between the Muslim and Christian philosophical traditions, and therefore he decided to expand his original research in that direction. In 1901, he published the first of a series of volumes that he had ambitions to write about Al-Ghazali (Asín Palacios 1901). The book figured as part of the *Colección de Estudios Árabes*, and featured a foreword by Menéndez Pelayo.

This was not the first title of the *Colección* to which Menéndez Pelayo contributed with a prologue, as he had already written a preface a year earlier for the first translation into Spanish of the famous twelfth-century philosophical novel by Ibn Tufail, *The Self-Taught Philosopher*, by another of Codera's disciples, Francisco Pons Boigues (1861–1899), which appeared posthumously in 1900 (Pons Boigues 1900). In the foreword to Ibn Tufail's novel, Menéndez Pelayo was not short of praise for the Andalusi philosopher, but at the same time he provided some less favorable judgments about the general body of Arabic literature. He credited *The Self-Taught Philosopher* as the richest work within "Hispanic-Arabic" literature and claimed that it was a source of embarrassment for Spain that there had not existed a Spanish translation before. The book deserved "to return, in Castilian, to the homeland of its author", and the Spaniards "should find glory in that [the author] was born in Spain, and neither his language nor his religion must impede us to count him as one of us" (Menéndez Pelayo 1900, pp. xi–xv) However, Menéndez Pelayo followed the tendency of previous scholars to "Hispanicize" the best outcomes of Andalusi culture and literature. He argued that the book was "barely Islamic" at heart, and indeed that it had few authentic Semitic elements.

In order to support such claims, Menéndez Pelayo argued that it was known that Ibn Tufail had belonged to a rationalist sect that was in constant tension with the Islamic orthodoxy, and which was Arab "only in the language". This rationalist school had flourished in areas of Muslim rule where the indigenous population, like in Syria and Spain, had inherited the cultural richness of pre-Islamic times. Ibn Tufail's novel had, according to him, a genuinely Spanish flavor, characterized by a "realistic idealism" and a "rooted sense of the self", which had saved the author from the "contemplative lethargy" that he would had received from the East. Menéndez Pelayo's essentializing criterion led him to conclude that such a work could not have been created in any other Muslim country than in al-Andalus, since it could have been produced only within "the Arabic civilization as it developed in our soil" (Menéndez Pelayo 1900, pp. xliii–xliv). Most interestingly, Menéndez Pelayo claimed to find parallelisms between the philosophical approach of Ibn Tufail and other Andalusi thinkers, and that of Spanish writers from the fifteenth to the seventeenth century, such as Baltasar Gracián (1601–1658). This prompted him "to suspect that there are laws that have not yet been uncovered, but that must be one day, which tie together the complicated historical relation of our forgotten philosophy over the centuries". Menéndez Pelayo thus came to assume in full the agenda of the Arabic school: "What is urgent today is to put at the hands of the scholars the principal documents in which the wisdom and thought of our elders is deposited, whether they were gentiles, Jews, Moors, or Christians, since the sun of science illuminated them all" (Menéndez Pelayo 1900, pp. xlvii–lxv).

None other was the scholarly purpose of "the most brilliant young Miguel Asín Palacios", as Menéndez Pelayo called him. Menéndez Pelayo therefore claimed to write "with authentic patriotic satisfaction" the foreword for Asín Palacios's work on Al-Ghazali in 1901. Asín Palacios's book, in his eyes, was to open "for the glory of Spain, a new path into the arduous and infrequent study of Oriental philosophy, and especially of Arab and Jewish philosophies, which interest us, Spaniards, in such a direct manner" (Menéndez Pelayo 1901, pp. vii–viii). The Catholic professor pointed out that both Pons Boigues, with the translation of *The Self-Taught Philosopher*, and Asín Palacios and Ribera with their texts about Llull and Ibn 'Arabi, had already initiated that renovation of Spanish scholarship in the history of ideas. But now Asín Palacios had embarked on a yet more intricate task by exploring the roots of Persian philosophy, "without which the origins and the development of our own philosophy would be unintelligible" (Menéndez Pelayo 1901, p. ix). Menéndez Pelayo's prologue challenged a common view that he thought most European Orientalists shared: that the works of Arab philosophy had been destroyed by the persecution of fanatic Islamic clergymen, and that they had only survived due to the

copies and translations made by Jews, who had transmitted them to the Christian public. Against that contention, Menéndez Pelayo differentiated between two major trends in Muslim medieval philosophy. On the one hand, there stood the works of peripatetic philosophers such as Averroes (1126–1198) and Avempace (1085–1138), who were "notoriously impious men", and whose works were suppressed by the zeal of Orthodox Muslims. But there existed as well philosophical works composed by God-fearing Muslim thinkers, such as the Sufis and the *mutakallim* (Muslim theologian) scholastics, which were still being printed and studied in the Muslim world, and which only recently had "receiv[ed] the attention of European Orientalists". This was something to celebrate, since Menéndez Pelayo argued that this constituted the most interesting part of Muslim philosophical thought. Not only was it marked by a cautious approach to rational inquiry that would not subvert the tenets of faith, but Al-Ghazali had been the master of those schools (Menéndez Pelayo 1901, pp. x–xiv).

After outlining some of the arguments of Asín Palacios, who characterized Al-Ghazali as a mystical thinker caught between the rationalist theories of the Muslim peripatetic philosophers and his own skepticism regarding the limitations of scientific thought, Menéndez Pelayo asserted that, ultimately, Al-Ghazali had been an apologist for reason's subservience to divine revelation, a detractor of rationalist philosophers for their subversion of religious life, and an adherent to the practical philosophy of asceticism. Menéndez Pelayo went on to claim that Al-Ghazali had been "the only thinker of his race who was able to exert a moral action over his coreligionists", and that because of his philosophical views, he "deserved to be a Christian". As Asín Palacios was to demonstrate, he said, Catholic science had made good use of Al-Ghazali's ideas and his calls to subordinate reason to religious life. The ascetic path Al-Ghazali had proposed was identical to the system that "is explained and recommended in the best devotional books used by Christian congregations" (Menéndez Pelayo 1901, pp. xx–xxiv). Al-Ghazali's teaching had been very influential among the Iberian thinkers of the three religions. Among the Muslims, his works had played a prominent role in the widespread diffusion of the Sufi school in al-Andalus. Among the Christians, Asín and Ribera had already pointed to connections between the mystic views of Ramon Llull and the works of Ibn 'Arabi. Menéndez Pelayo then endorsed the description made by the two Arabists of the Majorcan author as a "Christian Sufi", alleging that before Llull, the world of mystical speculation "had been inaccessible so far to Christian minds". Menéndez Pelayo also commended Asín Palacios's interpretation of Leo XIII's encyclical *Aeternis Patris* as setting the study of Islamic philosophy at the service and benefit of Catholic science. Nobody should be appalled, he argued, "by the singular history of a Muslim mystic who, over the centuries, has provided ammunition to the wisest vindicators of Christian dogma". On the contrary, "we should admire and imitate the magnanimous tolerance, the wide eclectic criterion" of those Christian scholastics who had put the works of Al-Ghazali to good use (Menéndez Pelayo 1901, pp. xxix–xxxix). Medieval Muslim philosophy was a legitimate source of authority for Catholic authors as long as it was useful for religious apologetics.

## 3. "Christianizing" Muslim Theology

In founding the *Revista de Aragón*, Asín Palacios established the perfect outlet for his research agenda into the study of Iberian Muslim philosophy and its impact on Christian philosophy and theology. In a series of articles about the Zaragoza-born Muslim philosopher Avempace, he adapted the nationalized interpretation of Andalusi culture to the Aragonese regionalism promoted by the journal (Asín Palacios 1900). Thus, he claimed that rarely had anybody suspected the fact that the city's name "resonated among peoples of a different race, a different language, and what is more, a different religion" (Asín Palacios 1900, p. 193). The Zaragozan philosopher had been influenced by other Muslim thinkers like the Persians Al-Ghazali and Avicenna, but, according to Asín Palacios, his was the exclusive credit for having introduced Oriental philosophy into Spain, where Muslim thought then reached new heights.

Asín Palacios's analysis of the Spanish Muslim school was rooted in his knowledge of the philosophy of Al-Ghazali. To him, the thread tying most Iberian Muslim thinkers together had been the

underlying weight that classical Greek philosophy had in all their intellectual systems. Al-Ghazali and Avicenna had been in his view the main channels through which that classical heritage had entered into Spain, despite Al-Ghazali's ambiguous stance toward rational philosophy. Al-Ghazali had engaged with Neo-Platonic philosophy, but had been wary of the dangers that free rational inquiry could have for religious life, and seemed to have been inclined to argue for an esoteric approach to rational speculation, limited to the few (Sedgwick 2017, pp. 41–43). Some personal notes of Asín Palacios reveal that at the time when he was composing his dissertation on Al-Ghazali, he was doubtful about the real, "intimate" attitude of Al-Ghazali toward rational philosophy.[5] For his part, Averroes had adopted a pantheistic system, developed by his master Avempace, which was essentially a mixture of the doctrines of Greek philosophers that assumed Neo-Platonist and mystical forms in the works of earlier Muslim philosophers like Avicenna and Al-Farabi (Asín Palacios 1900, p. 196). Although the importance that Avempace and Averroes had granted to the rational process distanced them from Al-Ghazali and his skepticism toward the virtues of science, they all constituted, through their works, a chain of knowledge connecting ancient classical culture and medieval Iberian philosophy.

　　Asín furthermore claimed that an alleged inherent distaste felt by Muslims for rational speculation had also gained these rationalist Muslim philosophers the hatred of the common people. The orthodox backlash against the medieval Muslim sects that had sought to make Greek thought compatible with Islam's core beliefs had caused the legacy of these rationalist philosophers to be proscribed, both in Spain and in the Orient. However, for some time, their works had been popular in Spain, and even reached followers of other religions. Asín Palacios therefore sought to recover the memory of those cultural and religious exchanges:

> The tolerance among men consecrated to the study of philosophy that can be appreciated in those medieval centuries, which many characterize as intolerant times, is a remarkable phenomenon. Muslims, Jews, and Christians, as they lived together and communicated peacefully in social commerce, except in the periods of political and religious warfare, so also did they cooperate in the quiet pursuit of truth. (Asín Palacios 1900, p. 301)

　　In 1902, Asín Palacios resumed the analysis of the philosophy of Al-Ghazali in order to familiarize the readers of *Revista de Aragón* with the doctrines of the Persian theologian on the question of religious belief (Asín Palacios 1902). He looked to highlight the resemblance between Al-Ghazali's doctrines and Christian theology on the topic of "faith". Behind his argument lay his avowed purpose of demonstrating that "the theological literature of the first centuries of Christianity" was at the root of Muslim thought (Asín Palacios 1902, pp. 386–92). Al-Ghazali's teachings had attempted to simplify Islamic law. To him, from a dogmatic point of view, the Muslim believer should be contented with merely knowing the Islamic profession of faith. It itself provided enough knowledge for being a good Muslim, and further theological pursuits should only take place in order to expel any religious doubts the believer held. The common people therefore must be guided by a simple, non-theological faith. They did not need to understand, only to believe. Theologians should then strive, Al-Ghazali argued, to keep the traditional sacramental formulas alive and ensure that most people abstain from reflecting on religious dogma, since according to him most men were like children in matters of faith.

　　The fundamental assumption of Al-Ghazali was that rational reflection was to be exclusively reserved for a small group of worthy devoted men, while the masses should find truth in a blind form of religious faith. Asín Palacios contended that Al-Ghazali's was not an original formulation, and that it must "be considered as a remnant of practices and doctrines that predate Islam, and which penetrated into Islam through ways that are yet unknown" (Asín Palacios 1902, p. 386). This opened new rich prospects of research. Asín Palacios, however, harbored few doubts that the sources of Al-Ghazali's views were located in the early Christian theological literature. He pointed to the condemnation of

---

5　Documentos de la Biblioteca Asín Palacios de la UNED, Caja "Notas e Ideas".

rational speculation made by Tertullian and the members of the Catechetical School of Alexandria as the most probable sources. Thus, Al-Ghazali's thesis was in fact "originally Christian", and as such, it reverberated, in its essence, "in all Catholic theologians". According to Asín Palacios, Pascal and Leibniz, for instance, had echoed the positions of the Persian theologian (Asín Palacios 1902, pp. 389–90).

The publications of the disciples of Codera in *Revista de Aragón* and their protagonist role in the editorial enterprise of the *Colección de Estudios Árabes* helped boost the image of the Arabist scholars as conforming to a homogeneous research agenda. The cohesion of the "School of Arabists" was strengthened by the organization of a volume of essays on Arabist themes that was arranged as a tribute to Codera as he retired from his position at the university in 1902. Asín Palacios would take the occasion of his homage to Codera to keep highlighting the contacts established between the philosophers of the three religions. This time, his text focused on the influence that the twelfth-century Islamic philosopher Averroes had over the theological views of the Christian scholastic theologian Thomas Aquinas (1225–1274). There had been, according to Asín Palacios, a very common misunderstanding that had led to considering Averroes to be taking sides with philosophy against revelation, and therefore that he was an enemy of Aquinas. But nothing was further from the historical truth. Averroes, "far from being the master and patron of Averroist rationalism, was indeed its most unyielding opponent" (Asín Palacios 1904a, p. 272). What Asín Palacios claimed was that the theories elaborated by both Averroes and Aquinas for conciliating faith and reason, rather than being opposed, were indeed identical. Their philosophical systems had to be compared in order to realize that the Christian saint had actually emulated the Arabic philosopher. Aquinas had established that there were supernatural truths that the rational mind could not know by their essence. However, reason was one of the gifts that God had given to mankind and, therefore, the theologian should not fear putting it to use for unraveling the mysteries of revelation. On the contrary, the theologian "should resolutely put philosophy inside the atrium of faith, as a guide that illuminates the path, as an assistant that helps him in the possible clarification of the mysteries, and as the battle weapon that defends him against error" (Asín Palacios 1904a, p. 277). Averroes had an analogous understanding, showing trust in the capacity of reason to gradually uncover the truth, but being at the same time aware that reason alone was impotent in unraveling the divine mysteries. Between the skepticism toward philosophy characteristic of the mystics, and the irreligious rationalism of the philosophers, both Averroes and Aquinas had opted for a middle way. Therefore, both had attracted the wrath of the traditionalists within their respective religions: Franciscan and some Dominican friars in the case of Aquinas, and Sufis, the Ulama, and the *mutakallimun* (theologians), in the case of Averroes.

Asín Palacios indeed went on to argue that the "Angelic Doctor" had directly emulated the Muslim thinker. He held that the Christian scholastic synthesis of the thirteenth century had to be "explained on the basis of the reception in Europe of the Muslim encyclopedia" (Asín Palacios 1904a, p. 308). The French philologist Ernest Renan (1823–1892) had argued that Christian scholasticism saw Averroes as a two-faced figure: as both the utmost impious philosopher, who had denounced the farce of the three religions, but also as the greatest commentator on the works of Aristotle. Asín Palacios claimed now to have demonstrated that Aquinas had only seen this latter aspect of Averroes. Asín Palacios then sought to elucidate the channels through which Aquinas could have had access to the thought of Averroes, indicating that Maimonides had likely been the main source, but pointing as well to Christian sources. The Dominican order of preachers, to which Aquinas belonged, had shown enthusiastic support for the study of science, and had been behind the creation of missionary schools under the patronage of the monarchs of Castile and Aragon for the study of Oriental languages in Tunisia and Murcia. Another Dominican friar of the thirteenth century, the Catalan Ramon Martí, had studied in those schools, and later composed several theological treaties intended to polemicize against Jews and Muslims, best known of which is *Pugio Fidei* (*The Dagger of Faith*). In Asín Palacios's eyes, Martí had borrowed extensively from the philosophy of Averroes in that work, and Aquinas would have most likely taken the doctrines of the Andalusi thinker from his Dominican coreligionist:

> And thus the scholastic synthesis incorporated the copious stream of the philosophy and even the theology of Averroes, purged from its errors against the Christian faith, as this theology was nothing else than an accommodation of the Christian dogma of the Oriental Church, adapted to the Islamic religion after an arduous and difficult gestation, made possible by the efforts of Al-Ghazali in the Orient, and by Ibn Tufail and Averroes in our Spain. (Asín Palacios 1904a, p. 324)

Drawing on the reception of the Eastern Christian adaptation of classical philosophy, Muslim thinkers such as Al-Ghazali and Averroes had developed a philosophical system that was to permit the harmonization of faith and reason, the ultimate goal of the contemporary movement for a Catholic science. Rather than seeing Al-Ghazali as a detractor of philosophy and Averroes as an impious rationalist, Asín Palacios praised both authors and their works as fundamental milestones in the formulation of the synthesis that Christian scholasticism would achieve between faith and reason, and which would later represent the theoretical underpinning for the apologetic works of the Neo-Thomistic revival. Spain had indeed played a fundamental role as a central location in the chain of these cultural transfers. The publications of Asín Palacios highlighting the encounters between Christian and Islamic thought gained him wide notoriety, a notoriety that was nevertheless not exempt from polemic. His articles riveted the attention of the ecclesiastical and the Catholic intellectual world on the findings of the Arabist school.

In 1901, his colleague in Zaragoza, Juan Moneva y Puyol (1871–1951), a professor on canon law, wrote a review in his volume about Al-Ghazali that defended Asín Palacios's approach to the subject from the point of view of a Catholic scholar. Speaking of Asín Palacios's clerical condition, Moneva y Puyol pointed out that "to some people, it is a rare combination to mix the priesthood with Arabic scholarship" (Moneva y Puyol 1901, p. 340). He claimed that for the common people, Arabism constituted a sort of affront to the Catholic Church, and that "eight centuries of *Reconquista* have nurtured such a level of hatred and distance that Moor and devil appeared to be similar things" (Moneva y Puyol 1901, p. 340). However, as Asín Palacios had demonstrated, the Muslim world was pervaded by Christian doctrines, and "the spirit of our Bible" imbued the Orient. The remnants of Greek thought that had existed in the East were channeled through the schools of the Eastern Christian philosophy. The Christian religion had flourished among the Arabs and its spirit had permeated "the scientific conscience of the Muslim people" (Moneva y Puyol 1901, pp. 341–42). The Quran, Moneva argued, contained many Christian ideas, and the doctrines of some Muslims philosophers were, for the most part, Christian, and even "identical to ours". In his opinion, Asín Palacios had demonstrated that this was the case with Al-Ghazali. Indeed, the Persian author could be invoked for the vindication of Catholic science. Al-Ghazali's arguments alleging the lack of conflict between reason and revelation stuck in the minds of the contemporary reader, at a time when "so many people devote themselves with great dedication to raising obstacles between science and faith, and to attack the Christian dogma all around". At times like this, Moneva would argue, the efforts of Asín Palacios were all the more valuable, as it had become more necessary than ever to resort to all possible means for the defense of religious faith. Moneva criticized the most rigorous Catholics who did not dare to consider worthy of their interest anything that was not inscribed within the narrowest Catholic intellectual tradition. Describing such attitude as an "empty nominalism", he went on to argue that if "we use electric light, the printing press, and machines that are all inventions made by heretics, and still we do not consider those inventions as wicked", likewise, "non-Christians can also argue about truths of faith" (Moneva y Puyol 1901, p. 342).

After *Revista de Aragón* disappeared and was replaced by *Cultura Española* (*Spanish Culture*), Asín Palacios would move his queries to the new journal. In 1906, he published a piece in which he claimed that the study of religious mysticism was relevant from the perspective of modern psychology (Asín Palacios 1906, pp. 209–35). Modern studies in analytical psychology were showing an increasing scientific interest in the "phenomena of the soul". According to this theory, in all times and places, mystics had sought divine truth in their inner intuitions, trained by ascetic lifestyle and meditational

practices. The mystics thus tended to pay a great deal of attention to the psychological processes of the mind and its operations. This was a very conspicuous characteristic of Al-Ghazali's works, as Asín Palacios had noted in his earlier studies of the Persian theologian, making him a good case study for modern psychological analysis. He proposed that Islamic mysticism, which emulated the ancient religious practices of Eastern Christianity, could prove to be of value for modern psychology, and its study could serve therefore to advance the goals of the movement for Catholic science in contributing to modern scientific production.

It is important to contextualize Asín Palacios' argument in concerns over the sequestration of Catholic scientific thought in this era. In 1903, Alberto Gómez Izquierdo (1870–1930), Asín Palacios's colleague at the section of philosophy of *Revista de Aragón*, wrote an article exposing some traits of Joseph Mercier's program for Catholic science at his institute for Thomistic philosophy in Leuven (Gómez Izquierdo 1903). In it, the Belgium cardinal was shown to decry the isolation in which Catholic men of science lived, not being able to reach out beyond their own small circles. Mercier's call to them was to cultivate science for science's sake, and by joining the scientific world on an equal footing with their non-Catholic peers, break away from the idea of Catholics as merely "soldiers" in the defense of their faith. That would be in his mind the best way to stress the compatibility between faith and reason. By stating the relevance of his Orientalist queries for modern psychology, Asín Palacios seemed to be following the advice of Mercier.

Already in 1904, Asín Palacios had informed the readers of *Revista de Aragón* with satisfaction that he had witnessed at the Second International Congress of Philosophy, convened that year in Geneva, the development of new scholarly trends that manifested "the emerging renewal of philosophy as a bulwark that reacts against positivism" (Asín Palacios 1904b, pp. 488–92). In his 1906 article on mysticism and psychology, he referred to a recent work by the leading American philosopher William James (1842–1910), dealing with the multiple forms in which the religious experience is manifested (James 2008). According to Asín Palacios, James's book had a powerful impact within the intellectual community in relation to the manner by which the phenomenon of mystical ecstasy was understood. Before James's publication appeared, the ecstasies described by the mystics had usually been addressed with derision and had not been considered as a subject worthy of scientific analysis. James had stirred a wave of new studies on that topic, but Asín Palacios complained that almost all of them focused on Christian mystical literature, which was not the only one available. He had then decided to enrich that trend with an analysis of Muslim mystical ecstasy, as described by Al-Ghazali and the Murcian Ibn 'Arabi. These authors not only represented two individual models, but also two genuine ideal types: Al-Ghazali embodyied a moderate and calmer approach, while Ibn 'Arabi represented a more "pathological" type of mysticism (Asín Palacios 1906, p. 210). Additionally, both authors had a significant impact on the religious practices of Islam, inspiring many sects and religious brotherhoods. In his analysis of the writings of both authors, Asín Palacios found that Al-Ghazali's were highly reminiscent of Christian Neo-Platonic doctrines. Indeed, he pointed out that such similarity of thought had forced Al-Ghazali to confess that, while the dogma of the Holy Trinity and the rejection of Muhammad were despicable aspects of the Christian faith, the rest of Christian doctrines in fact conveyed deep religious truths.

## 4. Conclusions

The nationalizing process of the Iberian Semitic legacy by Spanish intellectuals reached a peak at the turn of the twentieth century, as the "Arabist School" formed under the guidance and direction of Francisco Codera achieved its maturation. The visibility of Codera's disciples in journals such as *Revista de Aragón* and *Cultura Española* provided the Arabist field with an unprecedented level of notoriety among intellectual circles. As Julián Ribera and Miguel Asín Palacios "transcended" their master's reluctance to engage in the study of the "internal" history of the Andalusi legacy, they would put the analysis of the "Hispano-Muslim" philosophical output at the center of the queries of Spanish Arabism. Such a line of investigation had its roots in Ribera's interest during the 1890s in the history of the educational institutions of the Muslim world, prompted as it was by the public

debates in Spain about the nation's cultural and educational deficiencies. Taking Ribera's discovery of the influence of Islamic mystics on the obscure doctrines of Ramon Llull as a starting point, he, and especially Asín Palacios, would set themselves to unravel the previously unacknowledged points of intersection between Islamic thought and Christian scholasticism. This undertaking revealed the growing degree to which Orientalist scholarship in Spain was entangled with some of the most pressing intellectual debates of their time, particularly those polemics revolving around the role of religion in contemporary Spain.

The disciples of Codera manifestly tied their Arabist research to the goals of the movement for a Catholic science. It might seem paradoxical the fact that these overtly Catholic intellectuals were the scholars that went to great lengths in order to examine the philosophy and theology of Muslim Iberian thinkers and to reclaim their share in the Spanish intellectual tradition. However, their interpretation managed to impose a sort of "Christianization" on that legacy by means of stressing the Oriental Christian background in which those ideas were allegedly conceived. Such "Christianization" can therefore be understood as yet another stage of the nationalizing project of the Semitic legacy that Spanish Orientalism had put into practice for more than a century and that fitted to perfection the movement for a Catholic science. This new intellectual project of the "Arabist School" received the endorsement of the most renowned scholar of the Spanish conservative Catholic intelligentsia, Marcelino Menéndez Pelayo. He encouraged young Arabist scholars to further expand a line of inquiry that could prove very fruitful to his declared goal of bringing to light the grandeur of the Spanish intellectual tradition.

The arguments of Asín Palacios and Ribera, sustaining the thesis that medieval Muslim philosophy was in fact influenced by early Eastern Christian thought, neutralized to some extent the threat that Arabist inquiries could raise for the religious sensibilities of Catholic men of letters, and enhanced the capacity of the Arabist field to contribute to the development of Catholic science in Spain. In this sense, they emphasized the contribution made by Muslim philosophers and theologians to the harmonizing of faith and reason, which had a major impact on the scholastic synthesis of Aquinas. In light of the ambiguous approach toward modern science that characterized the Neo-Scholastic champions of the movement for Catholic science, such an undertaking proved useful in reframing the problem posited by modern science to Christian dogma in terms of the compatibility between reason and faith. The desire to engage with the larger scientific community was most evidently manifested in Asín Palacios's attempts to highlight the relevance of the study of religious belief and mysticism for modern scientific psychology. Ultimately, the young Arabist scholars would join other conservative Catholic colleagues in the call to partake in public efforts to modernize Spain's higher learning institutions and scientific production, constituting with that call a sort of conservative scholarly approach within Spanish regenerationism.

**Funding:** This research received no external funding.

**Conflicts of Interest:** The author declares no conflict of interest.

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
