# Peer review of "An Orientalist Contribution to “Catholic Science”: The Historiography of Andalusi Mysticism and Philosophy in Julián Ribera and Miguel Asín"

_religions, doi:10.3390/rel10100568_

Round 1

Reviewer 1 Report

This is a highly engaging and astutely considered article on the impact of Julián Ribera and Miguel Asín in which the author positions a very compelling analysis on these issues and contexts in Spain in stressing the significance of transnationalism and cultural/religious hybridity between Christians and Muslims.  

Author Response

Following the comments made by the second reviewer, I have made some revisions to my article. Most of them revolved around the goals pursued by the movement for Catholic science and its ambiguous stance toward the concept of science. The principal changes that has been introduced are to be found in lines 153-173, 193-222, 257-259, 387-391, 405-408, 548-561, 693-701, 740-764, 822-828.

Additional sources have also been included:

García Camarero, Ernesto and Enrique García Camarero. 1970. La polémica de la ciencia española. Madrid: Alianza. Gómez Izquierdo, Alberto. 1903. La escuela filosófica de Lovaina. Revista de Aragón IV: 403-8. Karamustafa, Ahmet T. 2007. Sufism: The Formative Period. Berkeley: University of California Press. Marín, Manuela. 2014. Reflexiones sobre el arabismo español: tradiciones, renovaciones y secuestros. Journal of Judaic and Islamic Studies, 1: 1- 17. 
 Menéndez Pelayo, Marcelino. 1881. Discursos leídos ante la Real Academia Española en la pública recepción del Doctor Don Marcelino Menéndez Pelayo el día 6 de marzo de 1881. Madrid: Imprenta de F. Maroto e Hijos. Menéndez Pelayo, Marcelino. 1887-89. La Ciencia española. Polémica, proyectos y bibliografía. Madrid: Colección de Escritores Castellanos, 3 vol. Motzkin, Gabriel. 1989. The Catholic Response to Secularization and the Rise of the History of Science as a Discipline. Science in Context, 3, 1: 203-26. Rivière, Aurora. 2000. Orientalismo y nacionalismo español. Estudios árabes y hebreos en la Universidad de Madrid (1843-1868). Madrid: Dykinson. Rodríguez de Cepeda, Rafael. 1897. Organización del Movimiento Científico Católico Contemporáneo. Valencia: Soluciones Católicas. Sedgwick, Mark. 2017. Western Sufism: From the Abbasids to the New Age. Oxford: Oxford University Press.

Reviewer 2 Report

This article scrutinizes the Spanish Orientalism of the late-nineteenth and early-twentieth century by contextualizing it against the backdrop of the Spaniards’ reconstruction of their own “national” identity as well as a “transnational” apologetic movement defending “Catholic science.” It focuses on two influential scholars, Julián Ribera (1858-1934) and Miguel Asín Palacios (1871-1944), the latter himself a Catholic priest, though other Orientalists associated with “The Arabist School”, such as Francisco Codera and Juan Moneva y Puyol, are also discussed. The two major arguments put forth are that these Orientalists tried to (1) “Hispanicize” the positive aspects of Andalusi culture and literature in order to integrate Islamic mysticism and philosophy into Spanish identity; and (2) utilize these aspects of their “Semitic past” in substantiating the claim that Catholic dogmas were compatible with scientific findings.

With a clear structure and fluent narrative, the article copes with an interesting yet under-studied issue: the relationship between Orientalism, European nationalism, and Christian apologism. It is quite successful in exhibiting the endeavors of the above Orientalists in incorporating medieval Muslim thinkers such as Al-Ghazali, Ibn Arabi, Averroes, and Avempace, into the Spanish “self” (first argument above). This integrating project is primarily carried out, as the article expresses, through disconnecting Islamic mystical and philosophical heritage from its Islamic domain and highlighting its roots in Eastern Christian philosophy and, through it, Hellenistic thought.

Besides these areas of strength, however, the article does not sufficiently elaborate its second key argument and does not clearly explain how this Hispanicizing project was utilized in defending Catholic science. For example, the text refers to Asín Palacios’ endeavor in contrasting the mystical irrationalism of Al-Ghazali with the rationalism of the Muslim philosophers Avempace and Averroes—who distanced themselves “from Al-Ghazali and his skepticism toward the virtues of science” (p. 9)—in order to absorb these two philosophers (Avempace/Averroes) into Spanish religious identity and strengthen its connection with science. Yet at the same time, Al-Ghazali’s irrationalism and his emphasis on inner intuition are considered in this article as “extremely relevant for modern psychological analysis,” since “the modern studies in analytical psychology were showing an increasing scientific interest in the ‘phenomena of the soul’” (p. 9). Therefore, integrating irrationalist Al-Ghazali into Spanish religious identity is also considered beneficial in connecting Christian dogma with science (psychology). Such seemingly inconsistent passages need to be reworked, and in particular the ambiguous term “science” (which is based on experiment/experience) and its connection to “philosophy” (based on reason) need to be defined and clarified.

As for the references, the article refers, aside from works by Asín Palacios and Ribera, to only ten sources, and only three of these were published after 1910. There are several updated, relevant works in the field which should be consulted, for example Mark Sedgwick’s recent monograph, Western Sufism, especially its first part “Premodern Intercultural Transfers” (between Neo-Platonism, Islam, Judaism, Christianity).

Concerning the structure, the section entitled “Engaging with the Internal History of al-Andalus” is too long (pp. 2-9), alone comprising more than the half of article. It is recommended that this lengthy section be divided into sub-sections, for instance, according to (a) background of the Arabist School (pp. 2-5); (b) Ribera and Asín Palacios (pp. 5-7); and (c) Menéndez Pelayo (pp. 7-9).

In addition to the above points, there are some minor issues to be addressed by the author:

Please mention the birth-death years after the names of major medieval and modern individuals mentioned in the text, including Asín Palacios and Ribera themselves (also in abstract). Please add the English translation for the title of Spanish works when they first appear in the article. Major figures need a brief description when they first appear: the reader is suddenly confronted with new names. It is better to use the complete family name “Asín Palacios” instead of “Asín,” since he is mainly known as “Asín Palacios” in English literature. Line 316: “eleventh century Al-Ghazali” should be replaced with “eleventh-twelfth century Al-Ghazali” (death 1111). Line 226: “thirteenth-century Majorcan theologian Ramon Llull” should be replaced with “thirteenth-fourteenth century…” (death 1315/16). Lines 260 and 280: please replace “Muhammad Ibn Arabi” with “Ibn Arabi.” Line 382: “Mutakallim” should be replaced with “mutakallim (Muslim theologian),” with “mutakallim” in italics. Line 501: “Mutakallim” should be replaced with “mutakallimun (theologians),” with “mutakallimun” in italics. Lines 530 and 532: “Puyol” is correct, not “Pujol” (twice).

Author Response

All corrections made by the reviewer have been incorporated, including birth-death years of authors mentioned in the text, the use of the complete name of Asín Palacios, and English translations of Spanish work titles. All the other minor issues identified at the end of the review have been fixed. The suggestion about the sections structured has been accepted as well, breaking the section entitled “Engaging with the Internal History of al-Andalus” into three smaller subsections, following the recommendation made in the review.

The reviewer’s comments regarding the weaknesses of the article with respect to the arguments relating to Catholic science has led me to reevaluate some of the points I raised in the article and to rework parts of it. I have specially reconsidered the ambiguous use of the term science throughout the article. In part I believe that this stems from the very ambiguity that characterized the movement for Catholic science. In the Spanish references to the movement that I used I have found very little efforts to theorize on the question of “science” besides a loose invocation to defend the compatibility of reason and faith. There was on the one hand a call to partake of scientific endeavors in all areas, which included natural sciences as well as humanities, but the major goal was clearly to combat some of the philosophical consequences that derived from positivist science, while vindicating that reason and rational inquiry could very well harmonize with religious faith if properly conducted. Asín Palacios viewed with satisfaction that some recent trends in philosophy were opposing positivism and were gradually more open to the study of phenomena related to spirituality, and in that sense I think that he raised the importance that the study of mysticism could have for modern psychology.

Regarding Al-Ghazali, some parts of the article has been reworked as to show that despite Al-Ghazali’s stance toward rational philosophy was highly ambiguous, the Arabist scholars emphasized his pivotal position in the chain connecting the reception of Greek philosophy in the Christian East and Western scholastics. He was therefore conceived to have been an important figure in the conformation of philosophical approaches seeking to harmonize faith and reason, as Moneva y Puyol is shown arguing in the last part of the article.

In light of all that was mentioned above, references to works of apologist of Catholic sciences has been added in order to clarify better the goals of the movement and the ways in which the Arabists sought to strengthen it. See particularly the revisions and additions made in lines 153-173, 193-222, 257-259, 387-391, 405-408, 548-561, 693-701, 740-764, 822-828.

As to the question of the limited number of references, an effort has been made to incorporate additional relevant sources. The work recommended by the reviewer of Mark Sedgwick has been very helpful, especially in broadening my understanding of the influence of Neo-Platonism into the different medieval theologies that are discussed in the article. A complete list of the references added includes the following:

García Camarero, Ernesto and Enrique García Camarero. 1970. La polémica de la ciencia española. Madrid: Alianza. Gómez Izquierdo, Alberto. 1903. La escuela filosófica de Lovaina. Revista de Aragón IV: 403-8. Karamustafa, Ahmet T. 2007. Sufism: The Formative Period. Berkeley: University of California Press. Marín, Manuela. 2014. Reflexiones sobre el arabismo español: tradiciones, renovaciones y secuestros. Journal of Judaic and Islamic Studies, 1: 1- 17. 
 Menéndez Pelayo, Marcelino. 1881. Discursos leídos ante la Real Academia Española en la pública recepción del Doctor Don Marcelino Menéndez Pelayo el día 6 de marzo de 1881. Madrid: Imprenta de F. Maroto e Hijos. Menéndez Pelayo, Marcelino. 1887-89. La Ciencia española. Polémica, proyectos y bibliografía. Madrid: Colección de Escritores Castellanos, 3 vol. Motzkin, Gabriel. 1989. The Catholic Response to Secularization and the Rise of the History of Science as a Discipline. Science in Context, 3, 1: 203-26. Rivière, Aurora. 2000. Orientalismo y nacionalismo español. Estudios árabes y hebreos en la Universidad de Madrid (1843-1868). Madrid: Dykinson. Rodríguez de Cepeda, Rafael. 1897. Organización del Movimiento Científico Católico Contemporáneo. Valencia: Soluciones Católicas. Sedgwick, Mark. 2017. Western Sufism: From the Abbasids to the New Age. Oxford: Oxford University Press.